# Risk Factors for COVID-19 in Inflammatory Bowel Disease: A National, ENEIDA-Based Case–Control Study (COVID-19-EII)

**DOI:** 10.3390/jcm11247540

**Published:** 2022-12-19

**Authors:** Yamile Zabana, Ignacio Marín-Jiménez, Iago Rodríguez-Lago, Isabel Vera, María Dolores Martín-Arranz, Iván Guerra, Javier P. Gisbert, Francisco Mesonero, Olga Benítez, Carlos Taxonera, Ángel Ponferrada-Díaz, Marta Piqueras, Alfredo J. Lucendo, Berta Caballol, Míriam Mañosa, Pilar Martínez-Montiel, Maia Bosca-Watts, Jordi Gordillo, Luis Bujanda, Noemí Manceñido, Teresa Martínez-Pérez, Alicia López, Cristina Rodríguez-Gutiérrez, Santiago García-López, Pablo Vega, Montserrat Rivero, Luigi Melcarne, María Calvo, Marisa Iborra, Manuel Barreiro de Acosta, Beatriz Sicilia, Jesús Barrio, José Lázaro Pérez Calle, David Busquets, Isabel Pérez-Martínez, Mercè Navarro-Llavat, Vicent Hernández, Federico Argüelles-Arias, Fernando Ramírez Esteso, Susana Meijide, Laura Ramos, Fernando Gomollón, Fernando Muñoz, Gerard Suris, Jone Ortiz de Zarate, José María Huguet, Jordina Llaó, Mariana Fe García-Sepulcre, Mónica Sierra, Miguel Durà, Sandra Estrecha, Ana Fuentes Coronel, Esther Hinojosa, Lorenzo Olivan, Eva Iglesias, Ana Gutiérrez, Pilar Varela, Núria Rull, Pau Gilabert, Alejandro Hernández-Camba, Alicia Brotons, Daniel Ginard, Eva Sesé, Daniel Carpio, Montserrat Aceituno, José Luis Cabriada, Yago González-Lama, Laura Jiménez, María Chaparro, Antonio López-San Román, Cristina Alba, Rocío Plaza-Santos, Raquel Mena, Sonsoles Tamarit-Sebastián, Elena Ricart, Margalida Calafat, Sonsoles Olivares, Pablo Navarro, Federico Bertoletti, Horacio Alonso-Galán, Ramón Pajares, Pablo Olcina, Pamela Manzano, Eugeni Domènech, Maria Esteve

**Affiliations:** 1Hospital Universitari Mútua Terrassa, 08221 Terrassa, Spain; 2Centro de Investigación Biomédica en Red de Enfermedades Hepáticas y Digestivas (CIBEREHD), 28029 Madrid, Spain; 3Hospital Gregorio Marañón and Instituto de Investigación Sanitaria Gregorio Marañón, Universidad Complutense de Madrid, 28040 Madrid, Spain; 4Hospital Universitario de Galdakao and Biocruces Bizkaia Health Research Institute, 48960 Galdakao, Spain; 5Hospital Universitario Puerta de Hierro Majadahonda, 28222 Madrid, Spain; 6Hospital Universitario La Paz and Instituto de Investigación Sanitaria La Paz (IdiPaz), Universidad Autónoma de Madrid (UAM), 28046 Madrid, Spain; 7Hospital Universitario de Fuenlabrada, 28942 Fuenlabrada, Spain; 8Department of Gastroenterology, Hospital Universitario de La Princesa, Universidad Autónoma de Madrid (UAM), 28006 Madrid, Spain; 9Instituto de Investigación Sanitaria Princesa (IIS-Princesa), 28006 Madrid, Spain; 10Hospital Universitario Ramón y Cajal, 28034 Madrid, Spain; 11Hospital Clínico San Carlos, Instituto de Investigación del Hospital Clínico San Carlos [IdISSC], 28040 Madrid, Spain; 12Hospital Universitario Infanta Leonor, 28031 Madrid, Spain; 13Consorci Sanitari de Terrassa, 08227 Terrassa, Spain; 14Hospital General de Tomelloso and Instituto de Investigación Sanitaria de Castilla-La Mancha (IDISCAM), 13700 Tomelloso, Spain; 15Hospital Clínic de Barcelona-IDIBAPS, 08036 Barcelona, Spain; 16Hospital Universitari Germans Trias i Pujol, 08916 Badalona, Spain; 17Fundación Hospital Universitario Doce de Octubre, 28041 Madrid, Spain; 18Hospital Clinic Universitari de Valencia, 46010 Valencia, Spain; 19Hospital de la Santa Creu i Sant Pau, 08025 Barcelona, Spain; 20Hospital Universitario Donostia and Instituto Biodonostia, Universidad del País Vasco (UPV/EHU), 20014 Donostia, Spain; 21Hospital Universitario Infanta Sofía, 28703 San Sebastián de los Reyes, Spain; 22Hospital Virgen de la Luz, 16002 Cuenca, Spain; 23Hospital del Mar and Institut Hospital del Mar d’Investigacions Mèdiques (IMIM), 08003 Barcelona, Spain; 24Complejo Hospitalario de Navarra, 31008 Pamplona, Spain; 25Hospital Universitario Miguel Servet, 50009 Zaragoza, Spain; 26Complexo Hospitalario Universitario de Ourense, 32005 Ourense, Spain; 27Hospital Universitario Marqués de Valdecilla and IDIVAL, 39008 Santander, Spain; 28Hospital Universitari Parc Taulí, 08208 Sabadell, Spain; 29Hospital San Pedro-Logroño, 26006 Logroño, Spain; 30Hospital Universitario y Politécnico de la Fe de Valencia, 46026 Valencia, Spain; 31Hospital Clínico Universitario de Santiago, 15706 Santiago de Compostela, Spain; 32Hospital Universitario de Burgos, 09006 Burgos, Spain; 33Hospital Universitario Río Hortega (HURH), 47012 Valladolid, Spain; 34Hospital Universitario Fundación de Alcorcón, 28922 Alcorcón, Spain; 35Hospital Universitari de Girona Doctor Josep Trueta, 17007 Girona, Spain; 36Hospital Universitario Central de Asturias and Instituto de Investigación Sanitaria del Principado de Asturias (ISPA), 33011 Oviedo, Spain; 37Hospital de Sant Joan Despí Moisès Broggi, 08970 Sant Joan Despí, Spain; 38Hospital Álvaro Cunqueiro, 36312 Vigo, Spain; 39Hospital Universitario Virgen de la Macarena, Universidad de Sevilla, 41009 Sevilla, Spain; 40Hospital General Universitario de Ciudad Real, 13005 Ciudad Real, Spain; 41Hospital Universitario de Cruces, 48903 Barakaldo, Spain; 42Hospital Universitario de Canarias, 38320 La Laguna, Spain; 43Hospital Clínico Universitario “Lozano Blesa” and IIS Aragón, 50009 Zaragoza, Spain; 44Hospital Universitario de Salamanca, 37007 Salamanca, Spain; 45Hospital Universitari de Bellvitge, 08907 L’Hospitalet de Llobregat, Spain; 46Hospital Universitario de Basurto, 48013 Bilbo, Spain; 47Consorcio Hospital General Universitario de Valencia, 46014 Valencia, Spain; 48Althaia Xarxa Assistencial Universitària de Manresa, 08243 Manresa, Spain; 49Hospital Universitario de Elche, 03203 Elche, Spain; 50Complejo Asistencial Universitario de León, 24071 León, Spain; 51Hospital Clínico de Valladolid, 47003 Valladolid, Spain; 52Hospital Universitario Álava, 01009 Álava, Spain; 53Hospital Virgen de la Concha, 49022 Zamora, Spain; 54Hospital de Manises, 46940 Manises, Spain; 55Hospital Universitario San Jorge, 22004 Huesca, Spain; 56Hospital Universitario Reina Sofía de Córdoba and Instituto Maimónides de Investigación Biomédica de Córdoba (IMIBIC), 14004 Córdoba, Spain; 57Hospital General Universitario de Alicante and Instituto de Investigación Sanitaria y Biomédica de Alicante (ISABIAL), 03010 Alicante, Spain; 58Hospital Universitario de Cabueñes, 33394 Gijón, Spain; 59Hospital Universitario Son Llàtzer, 07198 Palma, Spain; 60Hospital de Viladecans, 08840 Viladecans, Spain; 61Hospital Universitario Nuestra Señora de Candelaria, 38010 Santa Cruz de Tenerife, Spain; 62Hospital Vega Baja de Orihuela, 03314 Orihuela, Spain; 63Hospital Universitario Son Espases, 07120 Palma, Spain; 64Hospital Universitari Arnau de Vilanova de Lleida, 25198 Lleida, Spain; 65Complexo Hospitalario de Pontevedra, 36071 Pontevedra, Spain

**Keywords:** COVID-19, SARS-CoV-2, inflammatory bowel disease, 5-aminosalicylates, immunosuppression

## Abstract

(1) Scant information is available concerning the characteristics that may favour the acquisition of COVID-19 in patients with inflammatory bowel disease (IBD). Therefore, the aim of this study was to assess these differences between infected and noninfected patients with IBD. (2) This nationwide case–control study evaluated patients with inflammatory bowel disease with COVID-19 (cases) and without COVID-19 (controls) during the period March–July 2020 included in the ENEIDA of GETECCU. (3) A total of 496 cases and 964 controls from 73 Spanish centres were included. No differences were found in the basal characteristics between cases and controls. Cases had higher comorbidity Charlson scores (24% vs. 19%; *p* = 0.02) and occupational risk (28% vs. 10.5%; *p* < 0.0001) more frequently than did controls. Lockdown was the only protective measure against COVID-19 (50% vs. 70%; *p* < 0.0001). No differences were found in the use of systemic steroids, immunosuppressants or biologics between cases and controls. Cases were more often treated with 5-aminosalicylates (42% vs. 34%; *p* = 0.003). Having a moderate Charlson score (OR: 2.7; 95%CI: 1.3–5.9), occupational risk (OR: 2.9; 95%CI: 1.8–4.4) and the use of 5-aminosalicylates (OR: 1.7; 95%CI: 1.2–2.5) were factors for COVID-19. The strict lockdown was the only protective factor (OR: 0.1; 95%CI: 0.09–0.2). (4) Comorbidities and occupational exposure are the most relevant factors for COVID-19 in patients with IBD. The risk of COVID-19 seems not to be increased by immunosuppressants or biologics, with a potential effect of 5-aminosalicylates, which should be investigated further and interpreted with caution.

## 1. Introduction

Knowledge concerning the effect of the SARS-CoV-2 pandemic has grown exponentially. We previously published the largest cohort of patients with inflammatory bowel disease (IBD) and COVID-19 prospectively recruited with 12 months of follow-up in a national study [1]. A high percentage of patients in this cohort had occupational risk or were infected by intrafamilial transmission. We also confirmed that IBD does not worsen the COVID-19 prognosis, even with the use of immunosuppressants and biologics, as was shown elsewhere [2,3]. We demonstrated that COVID-19 affects neither the prognosis of IBD nor its treatment during the acute phase of infection or in the long term. In this sense, severe COVID-19 in patients with IBD is mainly related to older age [4,5] and comorbidities [2,6,7], as it occurs in the general population.

Investigations assessing factors that may favour the acquisition of COVID-19 in patients with IBD (case–control studies of patients with IBD with or without COVID-19) are scarce and needed [8]. Previous case–control studies either included COVID-19 patients from the general population as a control group [5], included a small proportion of COVID-19 cases or were retrospective cohorts [9,10]. The incidence of COVID-19 in patients with IBD is low [11]. This finding does not mean necessarily that patients with IBD are less susceptible to infection by SARS-CoV2, but could merely reflect special measures adopted in patients with a particular risk of infections [12]. Therefore, differences in factors that could influence the acquisition of COVID-19 must be assessed with the same type of population and under the same circumstances. This study complements the findings of our previous study [1] to define which characteristics favour or protect the occurrence of COVID-19 in IBD. Consequently, we sought to investigate, for the first time, two cohorts of patients with IBD, during the same time period and encompassing the first COVID-19 wave.

The present study aimed to assess epidemiological, demographic, and clinical factors that could influence the acquisition of COVID-19 in a large cohort of infected patients with IBD compared with noninfected patients.

## 2. Materials and Methods

### 2.1. Design

This study (COVID-19-EII study) was performed in the setting of the ENEIDA project, the Spanish registry of patients with IBD, promoted by the Spanish Working Group on Crohn’s disease and ulcerative colitis (GETECCU) [13]. ENEIDA is a prospectively maintained database that at the moment of study initiation had 60,512 patients with active follow-up (15 July 2020) in 86 hospitals. A total of 73 hospitals with 53,682 patients registered (89% of the entire database) accepted participation in this case–control study.

### 2.2. Study Population

Cases were all patients with COVID-19, diagnosed between March and July 2020 (during the first COVID-19 wave), who were identified by active search from their IBD unit (systematically addressing all the patients with IBD from each unit by email or phone call) or by direct notification from the emergency department, patient, family physician, or hospitalisation unit. Cases were matched with two controls (1:2) by age (±5 years), type of disease (Crohn’s disease [CD]/ulcerative colitis [UC]), sex, and centre. Both cases and controls came from the ENEIDA registry.

### 2.3. Definitions

A patient with IBD was considered a case if a COVID-19 diagnosis was made, based on a typical clinical picture that included fever (>38 °C), respiratory symptoms (cough and/or dyspnoea), dysgeusia or anosmia within the epidemiological setting. COVID-19 was confirmed by a positive diagnostic test including serology (IgM or IgG) or polymerase chain reaction (PCR) performed by nasopharyngeal swab for SARS-CoV-2. COVID-19 was considered probable in patients with a typical clinical picture but negative or lacking diagnostic tests.

The control group comprised patients with IBD without a COVID-19 diagnosis, coming from the same centre, during the study period (March–July 2020). The fact of not having COVID-19 was confirmed clinically by each attending physician through direct consultation with the patient, as at that time serologic or PCR tests were performed only for highly suspicious cases. Asymptomatic patients with IBD with positive PCR or serology were excluded from the study and were considered to have been infected with SARS-CoV-2 but without suffering from COVID-19.

Patients were considered to be in adequate compliance with the lockdown measures when they maintained social distance by staying almost exclusively at home since 14 March 2020, the date the Spanish government ordered a total lockdown to prevent the spread of SARS-CoV-2.

### 2.4. Data Collection

A prospective module hosted on the ENEIDA registry was specifically designed for this study to avoid missing cases and was externally monitored to ensure the correct acquisition of data. The data collected included clinical baseline characteristics such as date of IBD diagnosis and Montreal classification [14], type of IBD, family history of IBD, extraintestinal manifestations, and smoking behaviour at the time of infection. The following comorbidities were specifically registered, taking into account the moment of inclusion in the study as the time frame: chronic renal failure, dementia, chronic obstructive pulmonary disease, HIV, stroke, heart disease, congestive heart failure, diabetes mellitus, dyslipidaemia, arterial hypertension neoplasia, cirrhosis, rheumatological disease and immune-mediated disease, allowing the calculation of the Charlson comorbidity score [15]. These variables were collected at the time of the study (March–July 2022). Variables measuring the exposure risk to SARS-CoV-2 included occupational risk (such as health care workers, basic services such as supermarket cashiers, market clerks or pharmacy workers, teachers, workers of closed institutions, police and firepersons, animal control workers, veterinarians, or conservation and forest technicians), compliance with lockdown measures, social distancing and the route of contagion. The IBD therapeutic regimen was registered during the study period and included 5-aminosalicylates (5-ASA), systemic steroid treatment, immunosuppressants (cyclosporine, methotrexate, thiopurines, tacrolimus and tofacitinib) and biologics (anti-TNF, vedolizumab, and ustekinumab). All the controls were selected after the completeness of the inclusion period of cases (15 July 2020). They were identified by only one investigator blinded to the other characteristics to avoid selection bias.

### 2.5. Ethical Considerations

Written informed consent was obtained from all the subjects before inclusion in the registry. The Scientific Committee of ENEIDA approved the study on March 2020. It was also approved by the Ethics Committee of Hospital Universitari Mútua Terrassa (coordinating centre).

### 2.6. Statistical Analysis

Quantitative variables were correlated using the Mann–Whitney test for nonparametric data and Student’s *t* test for parametric data, while qualitative variables were compared using Fisher’s exact test or Chi2 test, when appropriate. Quantitative variables were compared using Student’s *t* test and the Mann–Whitney test, and the results were expressed as medians (± interquartile range (IQR) 25–75th percentiles).

Multivariable analysis was performed using conditional logistic regression analysis for case–control studies (COXREG in SPSS), where the presence of COVID-19 was the dependent variable. Variables with *p* value ≤ 0.1 on univariate analysis were introduced into the model. Because 5-ASA was more frequently used in patients with UC than in those with CD, the model was adjusted for UC diagnosis.

## 3. Results

Four hundred eighty-two cases with COVID-19 and nine hundred sixty-four controls without COVID-19 were included. The clinical characteristics of the patients with IBD and COVID-19 have been described previously in detail [1].

### 3.1. Clinical Baseline Characteristics

Table 1 shows the most important clinical characteristics of the cases and controls.

Cases less frequently had a family history of IBD [13% (64/482) vs. 25% (122/964); *p* = 0.02] and had a lower proportion of active smokers [11% (53/482) vs. 17% (160/964); *p* = 0.034] than the controls.

At least one comorbidity was observed in 43% (206/482) of cases and 36% (344/964) of controls (*p* = 0.01), with 24% (114/482) of cases and 19% (183/964) of controls having a moderate–severe Charlson score (score of three or more) (*p* = 0.02). The differences within specific comorbidities are shown in Figure 1 and Appendix A.

### 3.2. Epidemiological Risk Factors of Exposure

Table 2 summarizes the risk for COVID-19 related to occupational and epidemiological risk factors.

Almost one-third of cases (28%) and 10% of controls (*p* < 0.0001) were at risk because of occupational exposure. Health care professions were the most frequent occupational hazard [18% (85/482)] among cases and involved 4.5% (44/964) of controls]. One hundred thirty-three cases were considered to have an occupational risk. Among them, 96 (78%) were infected during their working activity. Most of these infections [70% (67/96)] occurred in the first month of the pandemic in Spain (March 2020).

Regarding other epidemiological risk factors for COVID-19, 44% of cases and 60% of controls (*p* = 0.154) declared good adherence to social distancing measures because of mandatory compliance, with no differences between groups. However, cases presented more preventive sick leave than controls [15% (73/482) vs. 8.5% (82/964); *p* < 0.0001], but a lower proportion of telecommuting [9.1% (44/482) vs. 19% (182/964); *p* = 0.009] and unemployment [3.3% (16/482) vs. 9.8% (94/964); *p* = 0.003]. Likewise, cases performed a worse overall total lockdown since the start of the state of alarm [50% (239/482) vs. 70% (676/964); *p* < 0.0001].

As stated previously, most of the cases were infected in March 2020, despite having had a higher proportion of preventive sick leave at the beginning of the pandemic [64% (47/73) in March 2020 vs. 36% (26/73) in April–July 2020; *p* = 0.037].

### 3.3. Treatment of IBD and COVID-19

Table 3 compiles the treatment received for both cases and controls for the period of time between March and July 2020.

No differences were found in the use of immunosuppressants and/or biologics. There was a tendency for greater use of steroids in cases than in controls [5.4% (26/482) vs. 3.6% (35/964); *p* = 0.06].

The use of 5-ASA was more frequent in cases [42% (204/482)] than controls [34% (332/964); *p* = 0.003]. However, a difference was found in the use of 5-ASA between patients with CD [14% (106/738)] vs. those with UC [61% (420/693); *p* < 0.001].

### 3.4. Risk Factors for COVID-19 in Patients with IBD

Considering the differences found in the univariate analysis (Table 1, Table 2 and Table 3, Appendix A), a multivariate model was conducted and is presented in Table 4.

A moderate (OR: 2.7; 95% CI: 1.3–5.9; *p* = 0.011) or severe (OR: 4.7; 95% CI: 1.7–12.7; *p* = 0.002) Charlson score, occupational risk (OR: 2.8; 95% CI: 1.8–4.4; *p* < 0.0001) and the use of 5-ASA (OR: 1.7; 95% CI: 1.2–2.5; *p* = 0.004) were independent risk factors for symptomatic COVID-19. Strict adherence to lockdown measures was the only factor protecting patients with IBD from contracting symptomatic infection for SARS-CoV2 (OR: 0.1; 95% CI: 0.09–0.2; *p* < 0.001).

Because the use of 5-ASA produced conflicting results regarding its potential effect on the association with COVID-19 acquisition and severity [16,17,18,19], we also used an adjusted model considering the Charlson index, corticosteroids, immunomodulators and biologics. The adjusted OR of the use of 5-ASA between cases and controls remained significant after adjusting for these additional factors: 1.8 (95% CI: 1.3–2.2; *p* < 0.0001).

## 4. Discussion

We showed that comorbidities and occupational risk were the two most relevant factors for having COVID-19 among patients with IBD in the ENEIDA cohort [13]. Thus, comorbidity is the most relevant risk factor as it is also related to harmful events due to COVID-19 [1,2,4,5,6,7]. Currently, data on the environmental exposure of noninfected patients are limited. Health care providers bore an enormous burden during the pandemic, as revealed in the data coming from Italy and China [20,21]. Almost one-third of cases and 10% of controls (*p* < 0.0001) had a job position considered to pose a high risk of infection and was the main cause for improper adherence to lockdown measures. Lockdown was demonstrated as the most effective strategy precluding SARS-CoV-2 expansion, also in patients with IBD [22]. Our study found that doing a strict lockdown was the only protective factor for COVID-19 in patients with IBD, regardless of their treatment.

We also confirmed that immunosuppressive treatment does not expose the patient to a greater risk of contagion. Univariate analysis revealed a trend toward greater use of steroids in patients with IBD who were infected (*p* = 0.06). The use of steroids is critical for other relevant infections in patients with IBD [12]. However, this finding was not replicated in the multivariate analysis (*p* = 0.203). On the other hand, 5-ASA, irrespective of the type of IBD, was the only drug independently related to symptomatic COVID-19 in patients with IBD.

The use of 5-ASA in patients with IBD and COVID-19 has shown conflicting results and is a debatable issue because the findings represent the first time that a nonimmunosuppressive treatment considered “safe” regarding infection risk was found to be potentially involved in both the acquisition of and/or having severe COVID-19 [23]. The first report of a possible harmful effect of 5-ASA on the COVID-19 outcome came from the first two publications of the SECURE-IBD registry, where associations of 5-ASA with hospitalisation and other severe COVID-19 outcomes were described, even after several and restrictive statistical analyses [17,24]. Nevertheless, the latest report [18] failed to show this relationship. The authors stated that the number of reported cases in the earliest assessments was too low to fully evaluate the association of 5-ASA in COVID-19 evolution. However, further analysis using a machine learning approach showed that 5-ASA was highly associated with COVID-19/IBD mortality [25], but this finding was not confirmed by another similar predictive model [26]. Considering this information, researchers of the SECURE-IBD registry were very cautious and concluded that the association of 5-ASA and COVID-19 was likely due to reporting bias and/or reporting delays [18].

By contrast, several other studies did not find 5-ASA to be related to COVID-19 [3,27]. One of the few IBD-COVID-19 case–control studies performed using a design similar to ours demonstrated that the rate of COVID-19 was similar between patients treated under therapeutic immunosuppression or users of 5-ASA compared to those who were not [9]. A recent case–control study demonstrated that SARS-CoV-2 has no impact on IBD clinical activity in patients under biological therapy, with no difference in the incidence of SARS-CoV-2 infection between users and nonusers of 5-ASA [10]. In two other studies of the Veterans Affairs Health Care System, neither thiopurines nor anti-TNF was associated with an increased risk of COVID-19 in IBD [2], and both vedolizumab and corticosteroids were independently associated with SARS-CoV-2 infection [19]. They reported no harmful COVID-19 events with the use of 5-ASA. However, 5-ASA was used as the reference treatment and was considered the “safest” [19].

Finally, a meta-analysis of 24 studies [28] showed that the risk of hospitalisation, intensive care unit admission and mortality due to COVID-19 in patients with IBD was higher only in patients using steroids or 5-ASA. This meta-analysis speculated that the increased risk of COVID-19 infection related to 5-ASA might reflect a confounding factor due to the use of 5-ASA as a proxy for an underlying UC. In our cohort, although 5-ASA was used differentially between patients with UC or CD, the effect of 5-ASA was independently associated with COVID-19, even when considering the diagnosis of UC in the model.

However, insufficient mechanistic data exist that link the use of 5-ASA and COVID-19. In fact, 5-ASA neither alters intestinal mucosal ACE-2 expression [29] nor changes SARS-CoV-2 infectivity [30]. On the other hand, 5-ASA exerts an anti-inflammatory effect in the colon by binding the peroxisome proliferator–activated receptor gamma (PPAR-γ). This receptor also increases the expression of ACE-2 and inhibits the expression of a transmembrane serine protease (TMPRSS2) relevant for viral entry with ACE-2 internalization and could be a potential mechanism of 5-ASA favouring COVID-19 [31].

Our study has several limitations. First, we did not collect information on the disease activity of the controls at the time of inclusion in the study. Overall, the use of immunosuppressive therapy between cases and controls in our current study was similar; thus, we can assume that there were no significant differences in this respect. Second, we included only symptomatic cases infected with SARS-CoV2. At the time of the first wave, neither PCR nor serology testing was mandatory and was not performed universally, raising the possibility of someone with asymptomatic or mildly symptomatic COVID-19 disease being considered a control. Finally, we must determine whether patients with IBD have a higher risk of contagion or a poorer COVID-19 outcome than the population of reference in population-based studies.

However, our study has crucial strengths. First, the study had national coverage with active participation of almost 90% of the IBD units from the Spanish ENEIDA registry. The universal access to health care within the National Health System in Spain [32] and adherence of most centres to the nationwide certification programme in IBD [33] provided homogeneity to this cohort. Additionally, although national case–control studies series on COVID-19 and IBD have been published previously [2,7,10], our study is the largest cohort of patients with IBD both with and without COVID-19.

## 5. Conclusions

In conclusion, we have shown that comorbidities and epidemiological risk factors are the most relevant aspects for COVID-19 in patients with IBD. We have also demonstrated that a strict lockdown is the only protective factor against the contagion. Finally, the potential impact of 5-ASA on SARS-CoV2 acquisition and COVID-19 severity should be confirmed or rejected in large-scale studies adjusted for potential effect modifiers and confounders to provide clear therapeutic recommendations. Presently, the risk–benefit of 5-ASA does not favour its withdrawal during the pandemic. Moreover, the findings of our study and others should be a warning not to assume certain apriorisms concerning the risk or harmlessness of certain drugs. The replacement of an effective therapeutic regimen in well-controlled patients by other drugs supposed to be less dangerous or total drug withdrawal may put patients at risk of severe decompensation.

## Figures and Tables

**Figure 1 jcm-11-07540-f001:**
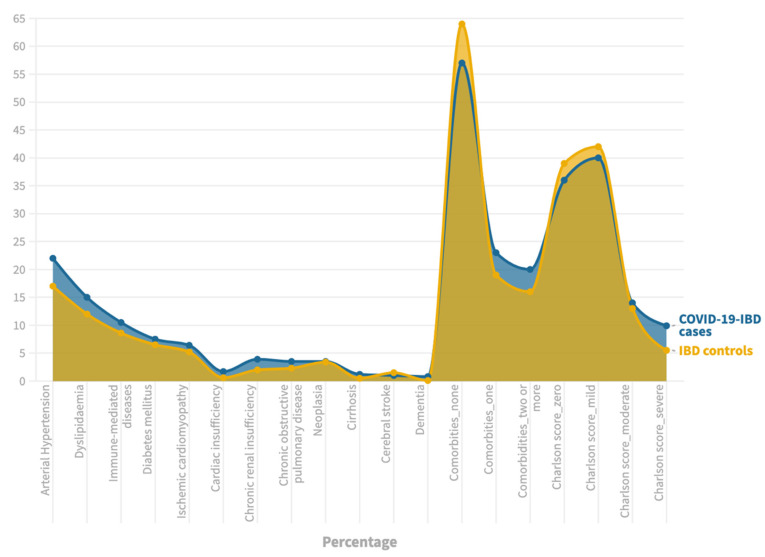
Comorbidities between cases and controls.

**Table 1 jcm-11-07540-t001:** Clinical baseline characteristics regarding inflammatory bowel disease between cases and controls. B1: inflammatory behaviour, B2: stricturing behaviour, B3: penetrating behaviour; L1: ileal, L2: colonic, L3: ileocolonic, L4: upper gastrointestinal tract; E1: proctitis; E2: left colitis; E3: extensive colitis. IQR: interquartile rate; IBD: inflammatory bowel disease.

Variable	Cases *n* = 482	Controls *n* = 964	Univariate *p*-Value
Male gender, *n* (%)	251 (52)	501 (52)	0.97
Age (years) Median (IQR)	52 (42–61)	53 (42–62)	0.75
Type of IBD, *n* (%)			0.60
Ulcerative colitis	221 (46)	471 (49)	
Unclassified colitis	14 (2.9)	1 (0.1)	
Crohn’s disease	248 (51)	492 (51)	
Crohn’s disease location, *n* (%)			
L1	114 (46)	232 (47)	0.79
L2	43 (17)	73 (15)	0.43
L3	88 (36)	186 (38)	0.48
L4 (isolated)	3 (1.2)	1 (0.2)	0.08
Crohn’s disease behaviour, *n* (%)			
B1	144 (58)	302 (31)	0.75
B2	71 (29)	147 (15)	0.72
B3	47 (19)	90 (9.3)	0.89
Perianal	59 (24)	164 (17)	0.85
Ulcerative colitis extent (%)			0.97
E1	43 (19)	85 (17)	
E2	80 (36)	183 (37)	
E3	98 (44)	203 (41)	
Extraintestinal manifestation, *n* (%)	125 (26)	247 (26)	0.98
Family history of IBD, *n* (%)	64 (13)	122 (25)	0.02
Smoking behaviour, *n* (%)			0.034
Active smoker	53 (11)	160 (17)	
Ex-smoker	137 (28)	261 (27)	
Never smoker	268 (56)	503 (52)	

**Table 2 jcm-11-07540-t002:** Lockdown and epidemiological risk factors between cases and controls.

Variable	Cases *n* = 482	Controls *n* = 964	Univariate *p*-Value
Occupational risk, *n* (%)	133 (28)	101 (10.5)	**<0.0001**
Healthcare	85 (18)	44 (4.5)	
Education	15 (3)	22 (2.3)	
Basic services (market clerks, supermarket cashier, pharmacy)	18 (3.7)	28 (2.9)	
Police and fireperson	5 (1)	2 (0.2)	
Closed institutions	2 (0.4)	2 (0.2)	
Veterinary, animal control worker or conservation and forest technician	4 (0.8)	3 (0.3)	
Social distance measures since the start of the state of alarm, *n* (%)	211 (44)	574 (60)	0.154
Sick leave	73 (15)	82 (8.5)	**<0.0001**
Retirement	60 (12)	191 (20)	0.348
Telecommuting	44 (9.1)	182 (19)	**0.009**
Unemployed	16 (3.3)	94 (9.8)	**0.003**
Others	18 (3.7)	25 (2.6)	**0.016**
Total lockdown since the start of the state of alarm, *n* (%)	239 (50)	676 (70)	**<0.0001**
Sick leave AND total lockdown since the start of the state of alarm, *n* (%)	73 (15)	82 (8.5)	
Sick leave WITHOUT total lockdown since the start of the state of alarm, *n* (%)	95 (19)	11 (1)	

**Table 3 jcm-11-07540-t003:** Differences in treatment for inflammatory bowel disease between cases and controls.

Variable	Cases *n* = 482	Controls *n* = 964	Univariate *p*-Value
5ASA, *n* (%)	204 (42)	332 (34)	**0.003**
Oral (oral and topic)	125 (26)	180 (19)	**<0.0001**
Topical (exclusive)	6 (1)	18 (1.9)	0.051
Monotherapy	142 (29)	214 (22)	0.9
Systemic steroids, *n* (%)	26 (5.4)	35 (3.6)	0.06
Immunosuppressants (all), *n* (%)	319 (66)	611 (63)	0.39
Immunosuppressants (in monotherapy), *n* (%)	113 (23)	191 (20)	
Azathioprine	90 (19)	160 (17)	
Mercaptopurine	8 (1.7)	7 (0.7)	
Cyclosporine	1 (0.2)	1 (0.1)	
Methotrexate	9 (1.9)	11 (1.1)	
Tacrolimus	1 (0.2)	2 (0.2)	
Tofacitinib	4 (0.8)	6 (0.6)	
Biologics (all), *n* (%)	235 (49)	493 (51)	0.28
Biologic (in monotherapy), *n* (%)	117 (22)	239 (25)	
Anti-TNF	71 (15)	134 (14)	
Vedolizumab	25 (5.2)	50 (5.2)	
Ustekinumab	21 (4.3)	52 (5.4)	
Combotherapy, *n* (%)	59 (12)	148 (15)	
Anti-TNF plus thiopurines	37 (7.7)	85 (8.8)	
Anti-TNF plus methotrexate	9 (1.9)	28 (2.9)	
Vedolizumab plus thiopurines	5 (1)	11 (1.1)	
Vedolizumab plus methotrexate	1 (0.2)	3 (0.3)	
Ustekinumab plus thiopurines	5 (1)	15 (1.6)	
Ustekinumab plus methotrexate	2 (0.4)	6 (0.6)	

**Table 4 jcm-11-07540-t004:** Risk factors for symptomatic COVID-19 in inflammatory bowel disease patients. IBD: inflammatory bowel disease. * The cases and controls were paired based on age, sex, disease type and hospital of reference.

Multivariate Analysis *		
Covariates	Adjusted Hazard Ratio (95% Confidence Interval)	*p*-Value
Family history of IBD	1.15 (0.9–1.5)	0.291
Active smoking	0.74 (0.4–1.04)	0.647
Charlson score		
Mild (one–two)	1.2 (0.7–2.2)	0.518
Moderate (three–four)	2.7 (1.3–5.9)	**0.011**
Severe (five or more)	4.7 (1.7–12.7)	**0.002**
Occupational Risk	2.8 (1.8–4.4)	**<0.0001**
Total lockdown since the start of the state of alarm	0.1 (0.09–0.2)	**<0.0001**
Systemic steroids	1.6 (0.8–3.1)	0.203
5-aminosalycilates	1.7 (1.2–2.5)	**0.004**
Ulcerative colitis	0.6 (0.08–5.2)	0.688

## Data Availability

The data underlying this article are available in the article and in its online Appendix A.

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
