# Peer review of "Risk Factors for COVID-19 in Inflammatory Bowel Disease: A National, ENEIDA-Based Case–Control Study (COVID-19-EII)"

_jcm, 2022, doi:10.3390/jcm11247540_

Round 1

Reviewer 1 Report

General Comments:

In this manuscript, Zabana et al. report the nationwide case-control study of patients with inflammatory bowel disease (IBD) with COVID-19. This study has strengths since it is the largest cohort of patients with IBD, both with and without COVID-19, covering almost all IBD patients in Spain. Also, it analyzed the details of the characteristics of patients with IBD, including medications, occupation, and social distance measures. Although there could be a discrepancy between the conclusion and the title/abstract, this manuscript contains significant findings for IBD during the covid-19 pandemic.

Major comments;

As the authors describe in the conclusions, the impact of 5-ASA on COVID-19 is still controversial, and the risk-benefit of 5-ASA does not favor its withdrawal during the pandemic. However, the title and abstract could have a high impact leading physicians and patients with IBD who read just the title and the abstract to withdraw 5-ASA. The title does not need to include the study results, and the sentences for the insufficiency of the 5-ASA risk, which is mentioned in the conclusion, should be added.

Reviewer 2 Report

No comments. It is a a well written, clear and honest case control research based on the prospective eneida registry exploring the risk of covid infection in ibd patient.

Reviewer 3 Report

Zabana Y et al conducted a 1:2 matched case-control study to assess the risk factors for COVID-19 infections for patients with IBD. The findings indicate that risk of COVID-19 was increased by 5-ASA not by immunosuppressants or biologics.

Major comments:

1.       Whether or not there is a causal relationship between 5-ASA and risk of COVID-19 should be stated with great caution. 5-ASA may be a mediator, not a true risk factor in the pathway. Causality cannot be drawn using this study design. Suggest using the word “association”, not “risk”. The title and conclusion in the Abstract read too strong. The results from the literatures are also mixed and controversial with regard to this matter. More UC patients than CD patients use 5-ASA. It is good that the authors matched disease type. However, 1) did authors assess 5-ASA as the only medication or combination with other medications? This needs to be clarified. 2) If patients are on 5-ASA alone, these might be mild IBD cases. It is likely that sicker patients take more protective measures (hard to be assessed in the study setting) that could influence the results.

2.       COVID is evolving rapidly. Data was collected in the first wave while vaccinations were not yet available. Results may change with varying variants and number of vaccinations received. We know that vaccination, wearing a mask, and social distancing may reduce infection. As the authors also pointed out that it was not possible to collect data on asymptomatic or mild cases or controls, would the cases be considered as a moderate or severe form of COVID? If so, this needs to be noted. Speaking of risk of COVID, severity is more concerning. For instance, it would be more interesting to know if a medication increases complications of IBD or COVID? Suggest adding more description about the study implications and how the findings can add to the exiting knowledge.

Other comments:

1.       Was dose of systemic corticosteroids measured?

2.       Was obesity included as one of the comorbidities?

3.       For the univariate p-values on Tables 1-3, what are the statistical methods used to incorporate the nature of the matched case and control sample?

Reviewer 4 Report

The errors in Table 4 bring into question the whole paper.  All the statistics presented need to be reviewed by the authors.

See the attached file for other comments

Round 2

Reviewer 4 Report

You have stated that differences of 42% vs 34% have a p value of 0.003.  Differences of 41% vs 33% from the same two samples have a p value of 0.27  You need to show the statistics in detail to defend this.

Author Response

REVIEWERS RESPONSE

(in the Word document the response includes figures)

Dear Emmanuel Andrès and Michael G. Hennerici

Editors-in-Chief

Journal of Clinical Medicine

November 20th, 2022

Dear Dr. Andrès and Prof. Hennerici

We are glad that the comments answered of our study “Risk factors for COVID-19 in Inflammatory Bowel Disease. A national, ENEIDA-based case-control study (COVID-19-EII)” (ref. number: jcm-1949196) to the reviewer 1, 2, 3 and 4 have been received accordingly and we really thank them as these changes the paper has improve very much.

However, we are sorry that reviewer 4 find our statistics inaccurate. We have reviewed in depth our work and we are truly thankful that we did. We deal to his/hers comments as follows (attaching results from our SPSS analysis when appropriated):

REVIEWER 4

Review revision for jcm-1949196

Title

Comorbidity and 5-aminosalicylates but not immunosuppression increases the risk of COVID-19 in inflammatory bowel disease: the COVID-19-EII study, a nationwide, ENEIDA-based, case–control study.

These have been dealt with satisfactorily:

Materials and Methods

Page 3 line 159-160 “60,512 patients actively controlled”.  The data base does not include controls.  This study is a case control study constructed from IBD positive data base subjects.  Please explain “actively controlled”.

Page 6 lines 239-242.  Please …..the cut offs for mild / moderate /severe here as well as in the table caption.

Overall : why was previous Montreal or some other IBD severity measure not included in the models, even if not up-to-date?

These have not been dealt with:

Results

Table 1 Crohn’s disease location – is that an exact chi square p value?  I should be due to the rarity of L4.  Also, P values should be given to 2 significant figures.

We have included the p values of each location (table 1)

Page 6 lines 239-242.  Please state the time frame used to calculate Charlton ……….

We have included the following phrase in the Methods section: “These variables were collected at the time of the study (March-July 2022).” Page 4, line 216

Page 7 lines 255-265.  This is confusing.  What is preventive sick leave?  If a subject must be sick to qualify, it cannot be used as a comparator.

As we stated in the previous revision: A preventive sick leave is the preventive withdrawal of working activity in patients with a particular risk. The main reason for this leave is the risk of infections, especially on immunosuppressed patients. So, the subject must not be sick to qualify, he/she must have to be considered to be in risk. Only that. This was a normal procedure with all immunosuppressed patients in Spain.

Tables 3

 first comparison 5ASA:  P=0.0039 from Fisher’s Exact Test so, if it must be rounded to one significant figure, it should be 0.004.

y

x

Frequency
Col Pct

1

2

Total

0

278
57.68

632
65.56

910

1

204
42.32

332
34.44

536

Total

482

964

1446

Fisher's Exact Test

Two-sided Pr <= P

0.0039

second comparison 5SAS oral and topic P=0.0020, not 0.27.  Which is incorrect, frequency or P value?

third comparison 5SAS topical should have P=0.27 (used above) not 0.051. 

fourth comparison 5SAS monotherapy should have P=0.042, not 0.9. 

Two comparisons presented as not significant may be significant.

The errors in Table 4 bring into question the whole paper.  All the statistics presented need to be reviewed by the authors.

Tables 3

first comparison 5ASA: P=0.0039 from Fisher’s Exact Test so, if it must be rounded to one significant figure, it should be 0.004.

Cases 204/ Controls 332.

Pearson Chi-Square 8,562, df 1, Asymptotic Significance (2 sided) 0.003 (not 0.004)

second comparison 5SAS oral and topic P=0.0020, not 0.27. Which is incorrect, frequency or P value?

We kindly thank the reviewer 4, as he/she pusses us to review every single data and we found a mistake here. There are 125 cases and 180 controls that used at the same time both oral AND topic 5-ASA

It is corrected on Table 3. Thank you indeed

 third comparison 5SAS topical should have P=0.27 (used above) not 0.051.

Thanks to the review of the second comparison, we found one extra case using topical 5ASA. However, this has no impact on the P value, as we are forced to use the continuity correction (p=0.051)

 fourth comparison 5SAS monotherapy should have P=0.042, not 0.9.

Again, thanks to the review of comparison 2, we have found more cases using 5ASA as monotherapy. However, this does not impact on the p- value

We are truly thankful to this reviewer persistence that forced us to review in depth our investigation. We have reviewed all statistic again and have not found any other inconsistency. Anyhow, if the reviewer agrees we can send the raw data so he/she can perform the statistics by him/herself.

REPETITION RATE

We have dealt with the repetition rate in this new version with a lot of effort as this is the second publication of the same protocol.

We feel that with these changes the paper has improve very much, and we hope that it will be now suitable for publication in Journal of Clinical Medicine.

Sincerely yours,

Yamile Zabana, MD, PhD

Hospital Universitari Mútua Terrassa & Centro de Investigación Biomédica en Red de Enfermedades Hepáticas y Digestivas (CIBEREHD).

Gastroenterology department and Inflammatory Bowel Disease Unit,

Plaça Dr Robert 5, 08221 Terrassa, Barcelona, Spain

Telephone: +34-937365050.

e-mail address: [email protected], [email protected]

Round 3

Reviewer 4 Report

I believe I missed this in my original review but Table 2 section Crohn’s disease location, n (%) had an incorrect P value.  The Mantel-Haenzel Chi square P=0.90, not 0.09.  The exact test should be used due to small cell size and P=0.93.   This section has been re-written in the current version and is no longer invalid. You cannot take a data set analysis with 3 degrees of freedom and divide it into 4 comparisons.

Table 3 5ASA section is now has new data with largely old P values.  When I plug the current data into a calculator I get P=0.0014 for oral (not < 0.0001); P=0.51 for topical (not 0.051) and P=0.0025 for monotherapy (not 0.90).

Author Response

REVIEWERS RESPONSE

Dear Emmanuel Andrès and Michael G. Hennerici

Editors-in-Chief

Journal of Clinical Medicine

November 26th, 2022

Dear Dr. Andrès and Prof. Hennerici

We are glad that the comments answered of our study “Risk factors for COVID-19 in Inflammatory Bowel Disease. A national, ENEIDA-based case-control study (COVID-19-EII)” (ref. number: jcm-1949196) to the reviewer 4 have been received accordingly and we really thank him/her as these changes the paper has improve very much. We have deal to his/her last comments as follows (attaching results from our SPSS analysis when appropriated):

REVIEWER 4

I believe I missed this in my original review but Table 2 section Crohn’s disease location, n (%) had an incorrect P value.  The Mantel-Haenzel Chi square P=0.90, not 0.09.  The exact test should be used due to small cell size and P=0.93.   This section has been re-written in the current version and is no longer invalid. You cannot take a data set analysis with 3 degrees of freedom and divide it into 4 comparisons.

We assume you are referring to Table 1 (where the location of Crohn’s disease is found). Now all the P-values are displayed for each location. However, the “global P-value” is in fact not 0.09, as previously stated, but 0.14. We introduce this data, as suggested by the reviewer, but keep the P-values of each location individually.

Global P-value

Table 3 5ASA section is now has new data with largely old P values.  When I plug the current data into a calculator I get P=0.0014 for oral (not < 0.0001); P=0.51 for topical (not 0.051) and P=0.0025 for monotherapy (not 0.90).

We have confirmed the data, and as expressed previous, our database is available if the reviewer wants to perform statistics with it.

We are truly thankful to this reviewer persistence that forced us to review in depth our investigation. We feel that with these changes the paper has improve very much, and we hope that it will be now suitable for publication in Journal of Clinical Medicine.

Sincerely yours,

Yamile Zabana, MD, PhD

Hospital Universitari Mútua Terrassa & Centro de Investigación Biomédica en Red de Enfermedades Hepáticas y Digestivas (CIBEREHD).

Gastroenterology department and Inflammatory Bowel Disease Unit,

Plaça Dr Robert 5, 08221 Terrassa, Barcelona, Spain

Telephone: +34-937365050.

e-mail address: [email protected], [email protected]
